# Immunological Response to Subcutaneous and Intranasal Administration of SARS-CoV-2 Spike Protein in Mice

**DOI:** 10.3390/vaccines12040343

**Published:** 2024-03-22

**Authors:** Mao Kinoshita, Kentaro Muranishi, Ken Kawaguchi, Kazuki Sudo, Keita Inoue, Hiroyasu Ishikura, Teiji Sawa

**Affiliations:** 1Department of Anesthesiology, Graduate School of Medical Science, Kyoto Prefectural University of Medicine, Kyoto 602-8566, Japan; ken1113@koto.kpu-m.ac.jp (K.K.); a080025@koto.kpu-m.ac.jp (K.S.); keitaino@koto.kpu-m.ac.jp (K.I.); anesth@koto.kpu-m.ac.jp (T.S.); 2Department of Emergency and Critical Care Medicine, Faculty of Medicine, Fukuoka University, Fukuoka 814-0133, Japan; muranishi@adm.fukuoka-u.ac.jp (K.M.); ishikurah@fukuoka-u.ac.jp (H.I.)

**Keywords:** SARS-CoV-2, subunit vaccine, adjuvant, prime and spike, humoral immune response

## Abstract

In novel coronavirus infection (COVID-19), the outbreak of acute lung injury due to trans-airway infection with the severe acute respiratory syndrome coronavirus 2 (SARS-CoV-2) is the starting point of severe disease. The COVID-19 pandemic highlights the need for a vaccine that prevents not only the disease but also its infection. Currently, the SARS-CoV-2 vaccine is administered via intramuscular injection and is generally not immunogenic to the mucosa. As a result, current vaccinations fail to reduce viral shedding and transmission and ultimately do not prevent infection. We established a mouse vaccine model in which a single dose of S1 protein and aluminum oxide gel (alum) subcutaneous vaccine was followed by a booster dose of S1 protein and CpG oligodeoxynucleotide intranasal vaccine. The group that received two doses of the intranasal vaccine booster showed a significant increase in IgG and IgA antibody titers against S1 and RBD in serum and BAL, and a significant difference in neutralizing antibody titers, particularly in BAL. One intranasal vaccine booster did not induce sufficient immunity, and the vaccine strategy with two booster intranasal doses produced systemic neutralizing antibodies and mucus-neutralizing antibodies against SARS-CoV-2. It will be an important tool against the emergence of new viruses and the next pandemic.

## 1. Introduction

Since its emergence in December 2019, Severe Acute Respiratory Syndrome Coronavirus 2 (SARS-CoV-2), the virus responsible for the COVID-19 pandemic [1,2], has profoundly affected human health and brought significant disruptions to the global economy. The rapid development and administration of multiple vaccines among humans have helped control the disease, reduce outbreaks, and decrease hospitalizations and deaths. There is also concern that the duration of the effect of the two doses of vaccine may be as short as 4 months, and the requirement for a third (booster) dose is being discussed in conjunction with the emergence of the next mutant strain [3,4]. Also, the emergence of SARS-CoV-2 variants of concern (VOCs) with increased infectivity, and the continued evolution of the virus have contributed to reduced vaccine efficacy [5,6,7]. Therefore, novel immune regimens and optimized vaccines that can control respiratory viral infections and induce long-term mucosal and systemic immune responses that can protect against different VOCs are necessary to prevent pandemics. Coronaviruses (CoVs) are enveloped viruses with a morphologically distinctive spike, which is spherical at the outermost point. The S protein contains two subunits, S1 and S2, with S1 containing the receptor binding domain (RBD), which is involved in the recognition and binding of host cell surface receptors, and S2, which is involved in fusing the viral and host membranes. S proteins are the target proteins of vaccines and neutralizing antibodies. SARS-CoV-2 virus can infect human respiratory epithelial cells via interactions with the human angiotensin-converting enzyme 2 (ACE2) receptor [8]. While research on the pathogenesis of novel coronavirus infection and vaccine antigens has progressed, several vaccine administration methods and combinations exist, and the associated effects, frequency, and duration of administration methods remain largely unknown. Our previous study has shown that the combination of S1 protein and CpG oligodeoxynucleotide (ODN) as an antigen and adjuvant combination, respectively, is effective as a subcutaneous vaccine [9]. The objective of this project is to elucidate the antibody titer transition in the combination of oral and intramuscular vaccines for novel coronavirus infection, and to develop an appropriate vaccine administration plan. Based on these results, we hypothesized that a single dose of conventional subcutaneous vaccine using S1 as antigen and alum as adjuvant and a booster dose of nasal vaccine using S1 as antigen and CpG as adjuvant would potentially induce nasal and mucosal immunity against novel coronavirus infection; we further designed animal studies to test this hypothesis.

## 2. Materials and Methods

### 2.1. Subunit Vaccines

The antigen utilized in this study was the S1 protein of SARS-CoV-2 (Spike S1-His [HPLC-verified], #40591-V08H, procured from Sino Biological, Beijing, China). To enhance the vaccine’s immunogenicity, aluminum hydroxide gel and CpG ODN were selected as adjuvants. The aluminum hydroxide gel used (InvivoGen, Alhydrogel, adjuvant 2%, #vac-alu-250, San Diego, CA, USA) was specifically formulated for subcutaneous administration. Additionally, the CpG K-type (K3) ODN (sequence: 5′-ATC GAC TCT GGA GGG TTC TC-3′) was synthesized by Gene Design in Ibaraki, Osaka, Japan, and was intended for intranasal administration.

### 2.2. In Vivo

ICR (Institute of Cancer Research) mice (5-week-old male, 25 g body weight) without protoplasts were prepared from Shimizu Laboratory Materials Co. (Kyoto, Japan). Approved by the Kyoto Prefectural University of Medicine Animal Experiment Committee in accordance with the Animal Welfare Law (#M2020-535, #M2021-337, and #M2022-328).

### 2.3. Immunization of Naïve Mice

The subunit vaccine was a recombinant S1 protein (0.5 mg/mL) adjuvanted with CpG-ODN or alum. The dose for subcutaneous administration was adjusted by dissolving the S1 (10 μg)-alum (1000 μg) or alum (1000 μg) to make a total of 200 μL. The dose for intranasal administration was S1 (10 μg)-CpG (1000 μg) or CpG (1000 μg) dissolved and adjusted to make a total of 30 μL. For subcutaneous administration, mice were immobilized for administration from the back. For intranasal administration, the mice were anesthetized with sevoflurane and 15 μL of subunit vaccine was administered into the right and left nasal cavity using a micropipette (30 μL total) [9]. The dose and method of administration of the subunit vaccine was as shown in Figure 1. Finally, mice were euthanized on day 56 and blood and bronchoalveolar lavage fluid (BAL) were collected. A series of experiments were independently carried out over several days to guarantee the reproducibility of the results.

### 2.4. Specimen Collection Methods

Blood samples were collected from either the tail vein or carotid artery under general anesthesia with sevoflurane. These samples were then centrifuged at 10,000 rpm for 10 min, after which the serum was separated and preserved at −80 °C in a solution containing 50% glycerol. Subsequent to euthanasia using sevoflurane, an incision was made in the neck’s epidermis to reveal the trachea, into which a catheter was inserted. Through this catheter, 2–3 mL of phosphate-buffered saline was gently infused, and the catheter was delicately aspirated 4–6 times. This procedure achieved a collection efficiency of approximately 70%. The aspirated fluid was then centrifuged at 1000 rpm for 10 min, and the supernatant was harvested, combined with 50% glycerol, and stored at −80 °C.

### 2.5. SARS-CoV-2-Specific Enzyme-Linked Immunosorbent Assay

The S1 protein (Sino Biologicals, SARS-CoV-2 Spike S1-His Recombinant Protein [HPLC-verified], #40591-V08H, Beijing, China) and the RBD protein (SARS-CoV-2 Spike Protein [RBD, His Tag], #40592-V08H, Beijing, China) were each diluted to a concentration of 2 μg/mL using a coating solution (0.1 mM NaHCO_3_, pH 9.6). These diluted antigens were then used to coat microtiter plates (430341, Nunc C96 Maxisorp; Thermo Fisher Scientific, Waltham, MA, USA), followed by incubation at 4 °C for 2 h. Afterwards, 200 μL/well of blocking solution was added, and the plates were further incubated at 37 °C either overnight or for 2 h. For the primary antibody, a 1000-fold dilution of mouse serum or bronchoalveolar lavage (BAL) fluid was applied at 100 μL/well and incubated overnight at 4 °C. Peroxidase-labeled anti-mouse IgG (Sigma-Aldrich, anti-mouse IgG [whole molecule] produced in goat-affinity isolated, buffered aqueous solution, #A4416, St. Louis, MO, USA) and anti-mouse IgA (Abcam, Waltham, MA, USA, goat anti-mouse IgA alpha chain [HRP], #ab97235) were subsequently used at a dilution of 1:60,000 for 1 h at 37 °C. Following six wash cycles, the plates were incubated with 2,2′-azinobis(3-ethylbenzthiazoline-6-sulfonic acid) (A3219; Sigma-Aldrich) at 25 °C for 30 min. Finally, 100 μL of 0.5 M H_2_SO_4_ was added to each well, and the optical density (OD) was measured at 450 nm using a microplate reader (MTP-880Lab; Corona Electric, Hitachinaka, Japan).

### 2.6. Receptor Blocking Antibody Responses

The SARS-CoV-2 Neutralizing Antibodies Detection Kit (#AG-48B-0002-KI01, AdipoGen Life Science, San Diego, CA, USA) was procured for the analysis. This kit utilizes antigens derived from SARS-CoV-2 (Wuhan-Hu-1 strain). Initially, the RBD protein, a key component of the kit, was pre-coated onto a 96-well plate. For the assay, the 100 μL/well of either 10-fold diluted serum or undiluted bronchoalveolar lavage (BAL) samples were added and incubated at 37 °C for 1 h. This step was followed by the addition of 100 μL/well of ACE2 (human)-HRP conjugate, which was also incubated for 1 h at 37 °C. Subsequently, 100 μL/well of the 3,3′,5,5′-tetramethylbenzidine (TMB) solution was introduced and incubated for 10 min in a dark environment. The reaction was halted by adding 100 μL/well of stop solution. Absorbance was then measured at 450 nm using a microplate reader (MTP-880Lab; Corona Electric, Hitachinaka, Japan). The quantification of neutralizing antibodies against SARS-CoV-2 in the serum/BAL samples was determined by calculating the percentage inhibition using the formula: (1 − (OD of the sample/OD of Negative Control)) × 100. The negative control comprised human serum diluted 1/10, confirmed to be negative for SARS-CoV-2 antibodies and screened for viral markers.

## 3. Results

### 3.1. Subcutaneously and Intranasally Administered Adjuvanted S1 Recombinant Protein Induces Humoral Immunity in Mice

#### 3.1.1. Evaluation of Humoral Immunity for Vaccination

On day 0, vaccine containing S1 (10 μg)-alum (1000 μg) or alum (1000 μg) alone was administered subcutaneously; on days 14 and 28, additional intranasal doses of vaccine containing S1 (10 μg)-CpG (1000 μg) or CpG (1000 μg) alone (n = 8/group). The group that received the subcutaneous administration of a vaccine containing S1 (10 μg)-Alum (1000 μg) and the intranasal administration of S1 (10 μg)-CpG (1000 μg) alone on days 0 and 28 was named Prime and Spike. The group that received the subcutaneous administration of a vaccine containing S1 (10 μg)-Alum (1000 μg) alone and the intranasal administration of S1 (10 μg)-CpG (1000 μg) on days 0, 14, and 28 was named Prime and Spike 2. Serum and BAL were collected on day 56 according to the vaccine administration schedule and IgG and IgA antibody titers were measured (Figure 1). Serum IgG titers against S1 and RBD in the Prime and Spike 2 group were significantly higher than those in the other groups (Figure 2a). A marked increase in the IgA antibody titer was observed in the serum against S1 and RBD only in the Prime and Spike 2 group (Figure 2b). However, additional experiments on serum IgA antibody titers against S1 and RBD showed no significant differences (Figure 2a,b).

As a result, the intranasal administration of Prime and Spike 2 significantly increased the IgA antibody titers against S1 antigen in BAL (Figure 3a). Significant increases in IgA antibody titers in BAL against RBD antigen were observed after the intranasal administration of Prime and Spike 2 (Figure 3b).

#### 3.1.2. Neutralization Titers against SARS-CoV-2

Based on the results of the above experiment, the reaction of neutralizing antibodies to the initial strain of SARS-CoV-2 (Wuhan-Hu-1) was measured and its presence was confirmed in serum and BAL. A significant increase in the number of neutralizing antibodies was observed. The neutralization titers in the Prime and Spike 2 groups were higher than those in the other groups (Figure 4). Neutralizing antibody titers for prime and spike were higher in the serum than in the adjuvant group, but not significantly different in BAL.

## 4. Discussion

This study was the result of a preclinical study to induce mucosal immune memory in the airways with a booster (spike) of adjuvanted intranasal inoculation using the immunity generated by primary vaccination (prime) in combination with subcutaneous and intranasal administration. First, we demonstrated that systemic neutralizing antibodies to SARS-CoV-2 and mucus-neutralizing antibodies to SARS-CoV-2 can be produced using S1 and CpG as a booster after the intranasal administration of the SARS-CoV-1 spike protein as an antigen in combination with alum adjuvant. In particular, antibody titers against S1 and RBD increased substantially after the two booster inoculations. The results in Figure 2 indicate that serum IgA levels against S1 and RBD were not elevated. Thus, systemic IgA elevation was less effective, and it is preferable to induce mucosal immunity using CpG as an adjuvant for the two-dose booster method. However, a single boost may not be sufficient to induce mucosal immunity. Therefore, two or more intranasal booster doses are required for mucosal immunity.

Using the SARS-CoV-1 spike protein as an antigen with alum and CpG as adjuvants and Prime and Spike’s new vaccine administration method combining subcutaneous and intranasal administration, we demonstrated that systemic neutralizing and mucus-neutralizing antibodies against SARS-CoV-2 can be produced. Recent studies have shown that the addition of the intranasal administration of mRNA-LNP in the absence of adjuvants induces a humoral immune response. In particular, immunoglobulin A is induced in the respiratory mucosa and systemic immunity is also enhanced. Thus, mice immunized with intranasal mRNA-LNP were found to be resistant to lethal SARS-CoV-2 infection. Intranasal vaccine prevents upper and lower respiratory tract infection [10]. The only respiratory mucosal vaccine currently approved is FluMist, which relies on attenuated live influenza virus. The attenuated influenza virus is sprayed directly into the nasal cavity to create a state of influenza pseudo-infection and induce immunity. By inducing secretory IgA in the nasal cavity, the entry of the influenza virus itself can be inhibited. Furthermore, the induction of secretory IgA antibodies into the mucous membranes of the whole body provides protection, and because the infection is similar to natural infection, the induction of CD8-positive T cells may prevent severe illness even if the virus strains are different. Currently approved SARS-CoV-2 vaccines for clinical use are all administered intramuscularly. These vaccines induce a serum IgG immune response but not mucosal IgA immunity [11].

These results suggest that a single boost may not be sufficient to induce mucosal immunity. Therefore, we believe that two or more intranasal boost doses are necessary for inducing mucosal immunity. Combinations with different administration sites could be widely applied as a new immunization strategy against emerging respiratory pathogens [12]. Alum has been used as an adjuvant in hepatitis B and human papillomavirus vaccines [13,14]. Fine particles, such as alum, activate inflammasomes and induce the secretion of proinflammatory cytokines in macrophages [15,16]. Alum, currently approved for clinical use by the U.S. Food and Drug Administration, causes cell death and the subsequent release of the host cell DNA, acting as a potent endogenous immune-stimulating signal [17]. The subcutaneous administration of S1-alums substantially increased serum IgG antibody titers against the S1 antigen and RBD antigen of spike protein S1 [9]. Furthermore, an increase in the serum-neutralizing antibody response has been observed [9]. Therefore, as a single dose of S1-alum prior to IN boosting does not seem to inhibit existing immunity but rather leverages it, we expect that IN boosting without an adjuvant would be equally or even more effective in individuals who have received multiple immunizations. In contrast, CpG ODN is a synthetic single-stranded DNA containing unmethylated CpG motifs that mimic genomic DNA from bacteria and viruses [18]. CpG ODN enhances innate and acquired immune responsiveness by activating the TLR9 signaling pathway in plasmacytoid dendritic cells (pDCs) and B cells. Synthetic oligonucleotides containing CpG motifs (CpG-ODN) have a stimulatory effect on the immune system and promote the maturation and activation of antigen-presenting cells, which promote the induction of Th1 and inflammatory cytokines [18,19]. Recently, CpG ODNs have been used as intranasal adjuvants. In influenza and tetanus toxoid vaccines, CpG ODNs activated immunity through mucosal administration. Multiple doses of recombinant SA1 protein from the intranasal route in combination with CpG adjuvant significantly increased the IgA antibody titers in serum and BAL fluid (Figure 3b). Furthermore, an increase in the response to neutralizing antibodies in the serum and BAL fluid was observed. Therefore, this drug may be suitable for booster administration to simultaneously induce mucosal immunity to combat newly emerging SARS-CoV-2 variants and potential pandemic SARS-like coronaviruses.

The intranasal vaccine could be used not only with the initial vaccination but also in combination with existing intramuscular vaccines to provide a booster effect after the second dose; its effectiveness was investigated in this study, including in combination with the initial intramuscular vaccination. In particular, the interval and number of boost doses are of great interest, and we were able to demonstrate the systemic neutralization against SARS-CoV-2 by using S1 and CpG as a boost vaccination after the subcutaneous administration of SARS-CoV-1 spike protein as an antigen in combination with alum adjuvant first, and then we were able to prove that the use of S1 and CpG as a boost inoculum produced systemic neutralizing antibodies and mucus-neutralizing antibodies against SARS-CoV-2. In particular, antibody titers against S1 and RBD increased significantly after the two booster inoculations. However, these results suggest that a single booster may not be sufficient to induce mucosal immunity. Therefore, two or more intranasal booster doses are required for mucosal immunity. If boost vaccination can be administered intranasally instead of subcutaneously, the need for medical personnel to administer the vaccine can be eliminated, the inoculum can be administered intranasally, and repeated doses can be administered more easily. Therefore, the vaccine is expected to be widely used to maintain specific immunity in the inoculated population, which will greatly contribute to the establishment of population immunity. Based on previous studies, we reported that (1) the subcutaneous administration of genetically recombinant RBD protein as an antigen does not activate the whole nasal immune system (Figure 2), and (2) the subcutaneous administration of genetically recombinant S1 protein can be performed as an antigen using CpG.

We previously reported that the intranasal administration of a vaccine using an ODN adjuvant activates the immune system in the airway mucosa. In a previously published paper, the subcutaneous administration of alum as an adjuvant with S1 as an antigen showed a significant increase in serum antibody titers against S1 and RBD after two doses, while antibody titers in BAL did not increase. As a result, the receptor-blocking antibody response in serum and BAL was low. Next, when S1 was administered intranasally as an antigen with CpG as an adjuvant, six doses were required. The results showed a significant increase in IgG and IgA antibody titers in serum and BAL against S1 and RBD. As a result, the receptor blocking antibody responses in serum and BAL were significantly high. These findings suggest that the combination of intranasal and intramuscular injections may increase antibody titers and neutralizing antibody responses to S1 and RBD [9]. Most currently approved vaccines require transmuscular administration, and the intervention of a healthcare worker is mandatory. Adverse reactions such as anaphylaxis immediately after vaccination and generalized adverse reactions such as muscle pain, fever, and general fatigue occur at a certain frequency, and these adverse reactions last for approximately one week, although they vary among individuals. The duration of the vaccine effect of booster vaccination is approximately 6 months at the earliest, and the necessity of such booster vaccination is being discussed in conjunction with the emergence of the next mutant strain [3,4]. The transmucosal vaccine targets the mucous membranes covering the eyes, mouth, and lungs, and by stimulating an immune response where the new coronavirus first enters the body, the mucosal vaccine can prevent infection and transmission to others, which is impossible with the current transmuscular vaccine. 

Some limitations of this study should be noted. First, the experiments focused on humoral immunity. However, vaccination in vivo did not allow us to examine how proteins secreted by immune system cells, such as cytokines, exert their physiological effects via specific receptors on the surface of target cells and what role they play in intercellular signaling. Second, there has been no vaccine efficacy testing or adverse event studies from the perspective of infection testing at the facility. Third, the use of SARS-CoV-2-specific Enzyme-Linked Immunosorbent Assay (ELISA) can detect neutralizing antibodies with a sensitivity of 82.9% and specificity of 98.3%. However, neutralizing antibodies cannot be quantified, suggesting that ELISA cannot completely replace the conventional neutralization test with virus-infecting cells in culture. The comparison between two commercial surrogate ELISAs to detect a neutralizing antibody response to SARS-CoV-2 [20]. Finally, we have not examined whether the vaccination method using the Wuhan strain has a neutralizing capacity against the various emerging coronavirus strains, particularly the Omicron variant. Assuming that these issues can be resolved through future research, the intranasal S1 protein with the CpG ODN vaccine can become an attractive option as an immune booster against virulent SARS-CoV-2.

## 5. Conclusions

Systemic and mucus neutralizing antibodies to SARS-CoV-2 can be induced by subcutaneous administration of S1 protein in combination with alum adjuvant followed by two boosted intranasal doses of S1 protein and CpG adjuvant.

## Figures and Tables

**Figure 1 vaccines-12-00343-f001:**
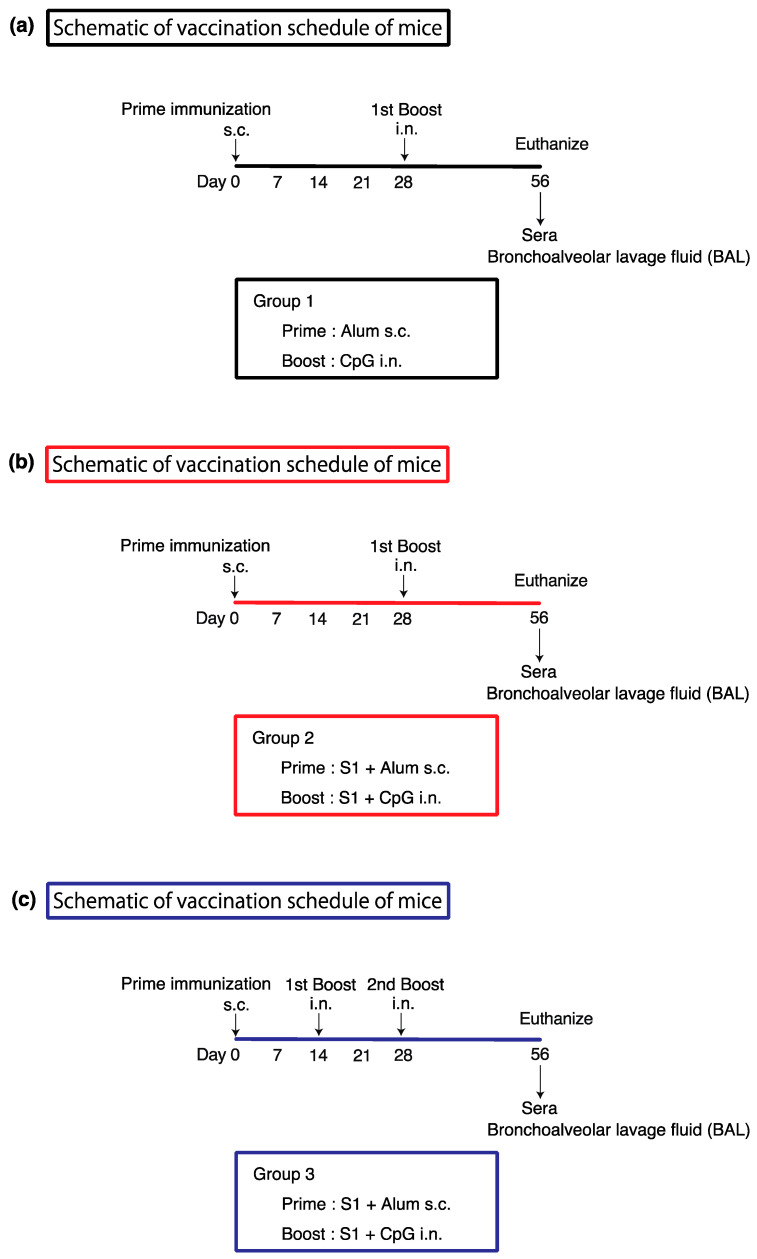
Vaccine administration protocol for mice. (**a**) Group 1: Mice initially received aluminum hydroxide gel alone (alum). This was followed 28 days later by a booster dose of CpG alone. Following the two scheduled immunizations, mice were euthanized for blood and bronchoalveolar lavage (BAL) fluid collection to measure the anti-S1 and anti-RBD antibody titers. (**b**) Group 2: Mice initially received either S1 protein combined with aluminum hydroxide gel (S1 + Alum). This was followed 28 days later by a booster dose of either S1 protein mixed with CpG oligodeoxynucleotide (S1 + CpG). Following the two scheduled immunizations, mice were euthanized for blood and bronchoalveolar lavage (BAL) fluid collection to measure anti-S1 and anti-RBD antibody titers. (**c**) Group 3: Mice were initially immunized with either S1 + alum. This was followed by booster doses of S1 + CpG on days 14 and 28. After the completion of the three scheduled immunizations, these mice were also euthanized for the collection of blood and BAL fluids to measure anti-S1 and anti-RBD antibody titers. Alum, aluminum hydroxide gel; CpG, CpG oligodeoxynucleotide; RBD, the recombinant receptor-binding domain of the S1 protein; S1, the SARS-CoV-2 spike S1-His recombinant protein.

**Figure 2 vaccines-12-00343-f002:**
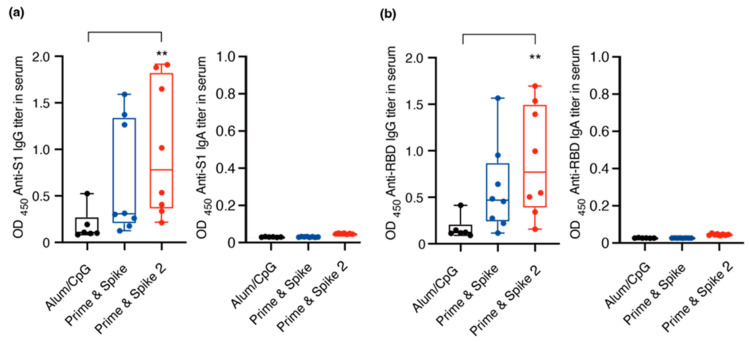
(**a**) Serum IgG and IgA antibody titers against the S1 protein, measured at a 1000-fold dilution. (**b**) Serum IgG and IgA antibody titers against the Receptor-Binding Domain (RBD), also at a 1000-fold dilution. The individual data points are represented by closed circles, while the median values are indicated by bars. A statistically significant difference, denoted by ** *p* < 0.01, was observed compared to all other groups. OD_450_ refers to the optical density measured at 450 nm. ‘RBD’ stands for Receptor-Binding Domain. In the ‘Prime and Spike’ group, participants received an initial subcutaneous administration of S1 protein combined with aluminum hydroxide (S1-alum) on day 0, followed by S1 protein mixed with CpG ODN (S1-CpG) on day 28. In the ‘Prime and Spike 2’ group, S1-alum was administered subcutaneously on day 0, followed by intranasal S1-CpG administrations on days 14 and 28. The absorbance values were compared using a one-way analysis of variance and the nonparametric Kruskal–Wallis test.

**Figure 3 vaccines-12-00343-f003:**
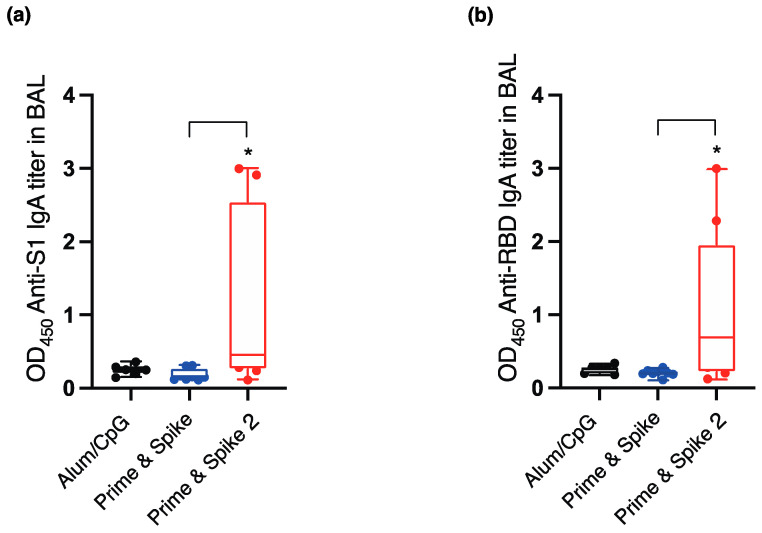
(**a**) Serum IgA antibody titers against S1 protein at a 1000-fold dilution. (**b**) Serum IgG antibody titers against the receptor-binding domain (RBD) at a 1000-fold dilution. Closed circles represent individual data points, while the bars indicate the median values. A significance level of * *p* < 0.05 was noted in comparison to all other groups. OD_450_ refers to the optical density measured at 450 nm. The term ‘RBD’ stands for Receptor-Binding Domain. In the ‘Prime and Spike’ group, the S1 protein conjugated with aluminum hydroxide (S1-alum) was administered subcutaneously on day 0, followed by the intranasal administration of the S1 protein with CpG ODN (S1-CpG) on day 28. In the ‘Prime and Spike 2’ group, participants received S1-alum subcutaneously on day 0, followed by intranasal S1-CpG on both days 14 and 28. To compare the absorbance values, a one-way analysis of variance and the nonparametric Kruskal–Wallis test were employed.

**Figure 4 vaccines-12-00343-f004:**
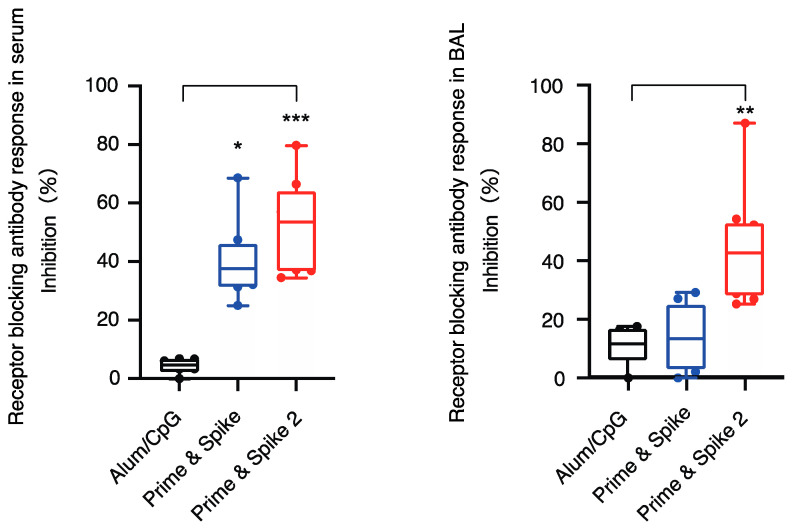
Serum and bronchoalveolar lavage (BAL) receptor blocking antibody responses to SARS-CoV-2 (Wuhan-Hu-1). Individual data points are depicted as closed circles, while the boxes and bars illustrate the mean ± standard deviation (SD). The significance levels are denoted as follows: * *p* < 0.05, ** *p* < 0.01, *** *p* < 0.001, indicating statistically significant differences when compared to the alum/CpG control groups. In the ‘Prime and Spike’ group, mice received an initial subcutaneous administration of S1 protein combined with aluminum hydroxide (S1-alum) on day 0, followed by an intranasal dose of S1 protein with CpG ODN (S1-CpG) on day 28. The ‘Prime and Spike 2’ group was similarly initiated with S1-alum subcutaneously on day 0, but received intranasal S1-CpG on both days 14 and 28. Statistical analysis of the inhibition data was performed using a one-way analysis of variance and the non-parametric Kruskal–Wallis test.

## Data Availability

The datasets generated and analyzed during this study and the raw data supporting the conclusions of this article are accessible from the corresponding author upon reasonable request.

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
