# Peer review of "Immunological Response to Subcutaneous and Intranasal Administration of SARS-CoV-2 Spike Protein in Mice"

_vaccines, 2024, doi:10.3390/vaccines12040343_

Round 1

Reviewer 1 Report

Comments and Suggestions for Authors

The paper is very interesting, and results can be used to improve vaccination schedules for obtaining optimal antiSARS-CoV-2 immunity.

Questions: 1) In fig.1 only 8 animals mentioned in each group. Is it enough for statistically verified conclusions. In materials and methods section it is mentioned that several experiments were conducted. Is these data available? Better to show it.

2) Did you study IgA levels in serum? It would be interesting to compare it with lavage levels.

3) In used neutralization test the only blocking of ACE binding to RBD is shown. Is it enough to conclude that obtained antibodies are neutralizing for the virus without conducting neutralization test with virus infecting cells in culture?

4) The main problem of antibodies to SARS-CoV-2 (generated by vaccination or monoclonal) is that they can loose neutralizing capacity for new mutated coronavirus strains. Did you study your vaccination method with Wuhan strain for different coronavirus strains, especially Omicron and other novel variants?

Suggestion for further studies - it would be interesting to study T-cell immunity development in your model.

Author Response

Questions: 1) In fig.1 only 8 animals mentioned in each group. Is it enough for statistically verified conclusions. In materials and methods section it is mentioned that several experiments were conducted. Is these data available? Better to show it.

[Response] We thank the reviewer for the careful reading. Significant differences were observed using the statistical software. However, due to the small sample size, all points were plotted and nonparametric methods were used. The graphs in Figures 2, 3, and 4 were revised to specify minimum and maximum values, median, and interquartile ranges instead of the mean and standard deviation, to better demonstrate the overall variability. Figure 1 was misleading and has been revised to clearly describe the grouping and immunization schedule. Group 1 consists of control mice sc-immunized with S1 alum, Group 2 consists of prime and spike, and Group 3 consists of prime and spike 2. Moreover, in the “Data Availability Statement” we have further specified that “raw data supporting the conclusions of this article” are accessible from the corresponding author upon reasonable request.

2) Did you study IgA levels in serum? It would be interesting to compare it with lavage levels.

[Response] Thank you for the opportunity to provide clarity here. Additional experiments on serum IgA antibody responses to S1 protein and RBD showed no significant differences. Figure 2 has been revised to present these findings.  

The following results and information have been included in the main text:

Page 4, Lines 165–167: However, additional experiments on serum IgA antibody titers against S1 and RBD showed no significant differences (Figure 2a and 2b).

Page 8, Lines 241–244: The results in Figure 2 indicate that serum IgA levels against S1 and RBD were not elevated. Thus, systemic IgA elevation was less effective, and it is preferable to induce mucosal immunity using CpG as an adjuvant for the two-dose booster method.

3) In used neutralization test the only blocking of ACE binding to RBD is shown. Is it enough to conclude that obtained antibodies are neutralizing for the virus without conducting neutralization test with virus infecting cells in culture?

[Response] We appreciate this insightful comment. For clarity, regarding the measurement of neutralizing antibodies, we have added the following statement to our discussion and cited the relevant reference.

Page 10, Lines 343–348: Third, the use of SARS-CoV-2-specific Enzyme-Linked Immunosorbent Assay (ELISA) can detect neutralizing antibodies with a sensitivity of 82.9% and specificity of 98.3%. However, neutralizing antibodies cannot be quantified, suggesting that ELISA cannot completely replace the conventional neutralization test with virus infecting cells in culture. Comparison of two commercial surrogate ELISAs to detect a neutralizing antibody response to SARS-CoV-2 [20].  

  1. Müller, K.; Girl, P.; von Buttlar, H.; Dobler, G.; Wölfel, R. Comparison of two commercial surrogate ELISAs to detect a neutralising antibody response to SARS-CoV-2. J Virol Methods 2021, 292, 114122. DOI: https://doi.org/10.1016/j.jviromet.2021.114122

4) The main problem of antibodies to SARS-CoV-2 (generated by vaccination or monoclonal) is that they can loose neutralizing capacity for new mutated coronavirus strains. Did you study your vaccination method with Wuhan strain for different coronavirus strains, especially Omicron and other novel variants?

[Response] Thank you for your valuable suggestion. The kit of neutralizing antibodies used in study for the Wuhan strain is also available for other coronavirus strains. However, we were unable to examine them in this study. Therefore, we have stated this as a study limitation and highlighted it as a perspective for further research. We have added the following statement to our discussion.

 Page 10, Lines 348–351: Finally, we have not examined whether the vaccination method using the Wuhan strain has neutralizing capacity against the various emerging coronavirus strains, particularly the Omicron variant.

Reviewer 2 Report

Comments and Suggestions for Authors

The manuscript “ immunological response to subcutaneous and intranasal administration of Sars-Cov-2 spike protein in mice” by M Kinoshita et al is a sort  of continuation of a paper recently published on Vaccine by these authors, using the same vaccine experimental approach  and readout but varying the immunization schedule.  The authors demonstrate the induction of  significant amounts of specific anti-S1 and anti -RBD IgG in the serum and IGA in BAL of vaccinated mice  by priming with subcutaneous administration of S1 protein in alum and boosting with S1-Cpg administered intranasally. Moreover, the induction of high amounts of mucosal IgA was demonstrated by a second in boost. These results are of interest in my opinion for Vaccine readers. However, the text needs major amendments and clarifications. 

In particular:

-       In the abstract the aim of the manuscript  and the achieved results should be more clearly summarized. In addition, in the discussion the relationship of this paper with the one previously published should be stressed as well as the importance of the modified vaccination schedule.

-       The introductory chapter final sentence (lines 57 -60) is wrong and not related to the reported results

-       Figure 1 is misleading . From the text I have understood that three groups of mice were analyzed in this manuscript. Group 1 should represent controls mice immunized sc with S1 alum ? ; group 2 should represent Prime & spike ;while group 3 Prime & Spike 2. If this is correct,  please better describe and clarify the groups composition and immunization schedules in this figure.

-       Figure 2 . In results ( line 157) is reported that this figure describe IgG amounts in serum and BAL. This is wrong as far as I understand. Data reported in figure 2 concern serum IgG  anti S1 and anti RBD

Author Response

-       In the abstract the aim of the manuscript  and the achieved results should be more clearly summarized. In addition, in the discussion the relationship of this paper with the one previously published should be stressed as well as the importance of the modified vaccination schedule.

[Response] Thank you for the encouraging feedback and valuable suggestion. In view of your comment, I have revised the abstract to clearly summarize the results obtained. Following is the revised abstract.

“In novel coronavirus infection (COVID-19), the outbreak of acute lung injury due to trans-airway infection with the severe acute respiratory syndrome coronavirus 2 (SARS-CoV-2) is the starting point of the severe disease. COVID-19 pandemic highlights the need for a vaccine that prevents not only the disease but also its infection. Currently, the SARS-CoV-2 vaccine is administered via intramuscular injection and is generally not immunogenic to the mucosa. As a result, current vaccinations fail to reduce viral shedding and transmission and ultimately do not prevent infection. We established a mouse vaccine model in which a single dose of S1 protein and aluminum oxide gel (alum) subcutaneous vaccine was followed by a booster dose of S1 protein and CpG oligodeoxynucleotide intranasal vaccine. The group that received two doses of the intranasal vaccine booster showed a significant increase in IgG and IgA antibody titers against S1 and RBD in serum and BAL, and a significant difference in neutralizing antibody titers, particularly in BAL. One intranasal vaccine booster did not induce sufficient immunity, the vaccine strategy with two booster intranasal doses produced systemic neutralizing antibodies and mucus-neutralizing antibodies against SARS-CoV-2. It will be an important tool against emergence of new viruses and the next pandemic.”

Furthermore, we have added the following statement to our discussion.

 Page 10, Lines 317–326: In a previously published paper, subcutaneous administration of alum as an adjuvant with S1 as antigen showed a significant increase in serum antibody titers against S1 and RBD after two doses, while antibody titers in BAL did not increase. As a result, the receptor-blocking antibody response in serum and BAL was low. Next when S1 was administered intranasally as an antigen with CpG as an adjuvant, six doses were required. The results showed a significant increase in IgG and IgA antibody titers in serum and BAL against S1 and RBD. As a result, the receptor blocking antibody responses in serum and BAL were significantly high. These findings suggest that the combination of intranasal and intramuscular injections may increase antibody titers and neutralizing antibody responses to S1 and RBD [9].

-       The introductory chapter final sentence (lines 57 -60) is wrong and not related to the reported results

[Response] Thank you for pointing this out. We apologize if the data conveyed such a remark. We have revised the following statement in the introduction.

Page 2, Lines 54–66: While research on the pathogenesis of novel coronavirus infection and vaccine antigens has progressed, several vaccine administration methods and combinations exist, and the associated effects, frequency, and duration of administration methods remain largely unknown. Our previous study has shown that the combination of S1 protein and CpG oligodeoxynucleotide (ODN) as an antigen and adjuvant combination, respectively, is effective as a subcutaneous vaccine [9]. Based on these results, we hypothesized that a single dose of conventional subcutaneous vaccine using S1 as antigen and alum as adjuvant and a booster dose of nasal vaccine using S1 as antigen and CpG as adjuvant would potentially induce nasal and mucosal immunity against novel coronavirus infection; we further designed animal studies to test this hypothesis.

-       Figure 1 is misleading . From the text I have understood that three groups of mice were analyzed in this manuscript. Group 1 should represent controls mice immunized sc with S1 alum ? ; group 2 should represent Prime & spike ;while group 3 Prime & Spike 2. If this is correct,  please better describe and clarify the grouping and immunization schedules in this figure.

[Response] Thank you for the careful reading. We apologize if the data conveyed such a remark.

Group 1 consists of control mice sc immunized with S1 alum, Group 2 consists of primes and spikes, and Group 3 consists of primes and spikes 2. We have modified Figure 1 to more clearly describe the grouping and immunization schedule.

-       Figure 2 . In results ( line 157) is reported that this figure describe IgG amounts in serum and BAL. This is wrong as far as I understand. Data reported in figure 2 concern serum IgG  anti S1 and anti RBD

[Response] Thank you for pointing this out. We apologize for the oversight. The data has been revised as follows.

Page 4, Lines 162–164: Serum IgG titers against S1 and RBD in the Prime & Spike 2 group were significantly higher than those in the other groups (Figure 2a).

Round 2

Reviewer 2 Report

Comments and Suggestions for Authors

In my opinion the manuscript has been sufficiently improved for publication.